



# Intermediate ions as indicator for local new particle formation

Santeri Tuovinen[1], Janne Lampilahti[1], Veli-Matti Kerminen[1] and Markku Kulmala[1]

[1]Institute for Atmospheric and Earth System Research, University of Helsinki, Helsinki, 00014,
Finland

*Correspondence to:* Markku Kulmala (markku.kulmala@helsinki.fi)

## Abstract

Atmospheric aerosol particles have considerable influences on climate via both aerosol-radiation
and aerosol-cloud interactions. A major fraction of global aerosol particles, in terms of their number
concentration, is due to atmospheric new particle formation (NPF) that involves both neutral and
charged clusters and particles. NPF is the major source of atmospheric intermediate ions, i.e.,
charged particles with mobility diameters between approx. 2 and 7 nm. We investigate ion
concentrations between 1.7 and 3.1 nm at the SMEAR II measurement station in Hyytiälä, Finland.
Both negative and positive ion number size distributions measured by Neutral cluster and Air Ion
Spectrometer (NAIS) are used. Our aim is finding the best size range of ions for characterizing local
intermediate ion formation (LIIF). Intermediate ion formation (IIF) refers to the formation of
intermediate ions through NPF, while local means that the growth of such ions from smaller clusters
has occurred in a close proximity to the measurement site, i.e., locally. We find that the ions in the
mobility diameter size range of 2.0-2.3 nm are the best suited for characterization of LIIF. The ion
concentrations in this size range indicate the elevated rates of IIF, and the potential distances the
growing ions have traveled are smaller than those for larger ions. In addition, in Hyytiälä, the
negative ion concentrations are more sensitive to IIF than the positive ion concentrations.
Therefore, we recommend the concentrations of ions with diameters 2.0-2.3 nm as the best choice
for characterization of LIIF.

## 1 Introduction

Atmospheric aerosol particles affect climate on local, regional and global scales (Boucher et al.,
2013; Rosenfeld et al., 2014; Quaas et al., 2022; IPCC, 2022). These particles scatter radiation,
impacting Earth's radiative balance (Bellouin et al., 2005; Yu et al., 2006). In addition, particles
with diameters larger than about 50-100 nm are able to act as cloud condensation nuclei (CCN)
(Komppula et al., 2005; Anttila et al., 2010; Bougiatioti et al., 2020). CCN are a necessity for cloud
droplet formation, and CCN number and properties influence cloud properties such as cloud
irradiance (Rosenfeld et al., 2014; Fan et al., 2016). A large fraction of the global aerosol number
concentration is due to atmospheric new particle formation (NPF) (Spracklen et al., 2010; Gordon et
al., 2017).

During NPF, sub-2 nm atmospheric aerosol particles are forming by gas-to-particle conversion,
after which they start growing to larger sizes (Kulmala et al., 2001; Kerminen et al., 2018).



Eventually, the particles created due to NPF might reach sizes, where they can have impacts on e.g., climate or air quality. We consider the growth of the particles as a necessary prerequisite for NPF. Therefore, even if small molecular clusters are forming, but there is no growth, or the growth is negligible, we do not consider NPF having taken place.

So-called NPF events, during which the formation and growth of particles is seen, are regularly observed all over the globe, from boreal forests to urban megacities (Dal Maso et al., 2007; Dada et al., 2017; Kerminen et al., 2018; Chu et al., 2019; Bousiotis et al., 2021; Brean et al., 2023). In addition, there is so-called quiet NPF, taking place outside the traditional NPF event times (Kulmala et al., 2022). NPF has been observed to occur regularly at the SMEAR II measurement station in Hyytiälä, southern Finland, (Dal Maso et al., 2005; Nieminen et al., 2014; Dada et al., 2017). Over 20% of the days in Hyytiälä are classified as NPF event days (Dada et al., 2018), during which NPF often occurs on a regional scale. In addition, local evening and nighttime clustering events have been observed (Mazon et al., 2016). The days classified as NPF event days in Hyytiälä have been estimated to contribute a majority to the particle production, while quiet NPF is responsible for around one fifth of the particle production (Kulmala et al., 2022). Furthermore, NPF contributes significantly to CCN production (Sihto et al., 2011).

The extent of particle production due to NPF depends on environmental conditions. For example, low levels of particle pollution and sufficient abundance of potential precursor vapors, such as sulfuric acid, bases and oxidized organic compounds, tend to favor NPF (Paasonen et al., 2010; Kulmala et al., 2013a; Schobesberger et al., 2013; Dada et al., 2017; Kerminen et al., 2018; Lehtipalo et al., 2018; Yan et al., 2021). Therefore, for example during any regional-scale NPF event, different local environments within the region of interest are expected to provide different contributions to the regional new particle production. To accurately evaluate the strength of the local particle production, the influence of particles originating from outside the area of interest should be minimized.

In previous studies (Hõrrak et al., 2000; Hirsikko et al., 2005; Kulmala et al., 2007; Virkkula et al., 2007; Hirsikko et al., 2011), atmospheric clusters, referring to particles with mobility diameters smaller than approximately 2 nm, have been observed to exist all the time, as predicted by Kulmala et al., 2000. The majority of these clusters are neutral, however a fraction of them are charged ions (Kulmala et al., 2007). Due to the large number of ever present neutral clusters, and ionization due to e.g., cosmic and gamma radiation and radon decay, the concentrations of atmospheric ion clusters are relatively stable (Laakso et al., 2004; Tammet et al., 2006). Therefore, as we consider the growth of particles a prerequisite for NPF, we cannot detect NPF reliably from the concentrations of neutral or charged clusters because such concentrations do not tell us whether the clusters are growing or not in size. We note that based on the measured number concentrations of sub-2 nm ions, it is not possible to separate large charged molecules from charged molecular clusters. Therefore, cluster ions and charged molecules are from hereon referred to together as small ions.

In contrast to small ions, concentrations of intermediate ion (ions with mobility diameters approximately between 2 and 7 nm) have been observed to be very low except during periods of



atmospheric new particle formation, rain, snowfall or snowstorms (Hõrrak et al., 1998; Hirsikko et al., 2007; Hirsikko et al., 2011; Tammet et al., 2014; Leino et al., 2016). During NPF, intermediate ions are being formed through ion mediated nucleation pathways or through the attachment of small ions with neutral particles. Therefore, increased concentrations of intermediate ions can be considered indicative of the occurrence of NPF (Tammet et al., 2014; Leino et al., 2016).

In this work, we will investigate the use of atmospheric intermediate ion concentrations for studying local NPF. There are two important issues connected with this. First, we want to observe ions that have already started to grow to larger sizes, and can thus be connected to NPF as per our definition. Second, the activation of clusters for growth should occur as locally (on site) as possible. We will refer to the formation of intermediate ions as IIF (intermediate ion formation), and the local formation of intermediate ions as LIIF. The separate term for intermediate ion formation compared to NPF is used to make it clear that we are observing and studying the formation of charged ions. To what extent the formation of neutral particles is taking place at the same time, is not known.

The concentrations of intermediate ions can be affected by transported ions, i.e., ions, which have traveled with moving air masses during their growth. Because of this, the factors which have lead to the activation of the clusters for growth might differ from those for the ions originating from closer to where they are detected. In this study, our aim is to use ion concentrations to characterize LIIF, and thus we hope to minimize the influence of transported ions.

In this study, we will investigate intermediate ion concentrations measured in Hyytiälä, Finland, using a Neutral cluster and Air Ion Spectrometer (NAIS) (Mirme and Mirme, 2013; Manninen et al., 2016). Our aim is to find out the optimal size range of intermediate ions to be used in characterization of LIIF. In addition, both ion polarities will be compared, and the potential impact of polarity on intermediate ion concentrations, and therefore on the characterization of LIIF, will be evaluated. The potential contribution of transport on the ion concentrations will be discussed. Finally, a recommendation for the best ion diameter to use in the characterization of LIIF with minimal influence from transported ions is given.

## 2 Methods

### 2.1 Ion number size distribution data

We used ion number size distribution data from SMEAR II (Station for Measuring Forest Ecosystem–Atmosphere Relations II) measurement station (Hari and Kulmala, 2005). SMEAR II station is located in Hyytiälä, southern Finland (61°51´N, 24°17´E, 180 m). The site is surrounded by a relatively homogeneous Scots pine forest. For more details on the site and the measurements therein, see e.g., Manninen et al. (2009b) and Nieminen et al. (2014).

The used ion number size distribution data were measured with a NAIS (Neutral cluster and Air Ion Spectrometer) (Mirme and Mirme, 2013; Manninen et al., 2016). The NAIS is able to measure both




air ions (mobility diameters 0.8–42 nm) and total particles (mobility diameters 2.5–42 nm) by the use of a corona charger. The data were inverted using the v14-lrnd inverter (Wagner et al., 2016). The time resolution of the data was two minutes. The measurement height for the NAIS measurements is 2 meters. Due to the presence of charger ions in diameters up to 2.5 nm in the total

120 particle size distributions measured by the instrument (Manninen et al., 2009b; Mirme and Mirme, 2013), we restricted our analysis to ions in this study.

The ion size number distribution data were used from between 4th of January 2016 and 31st of December 2020. The data coverage was good for the whole period, with few major gaps of more than 24 hours in the data.

## 2.2 Ion number concentration analysis

The data were used from all the available days, and no distinction was made based on whether the days had been classified as NPF days or not. Recent advances have shown that NPF does occur even during the days classified as non-event days (Kulmala et al., 2022). As such, we chose to include all the days in the analysis, regardless of whether a NPF event was observed to occur during

the day, or not.

Four different ion size bins, which were based on the used inversion method, were considered in the analysis (Table 1). The ion concentration values equal or below zero were omitted, and outliers were removed based on the 1% and 99% quantiles. However, we note that the effect of this procedure on our results was found to be minor.

Median, 25%, and 75% quantile concentrations were determined for each hour of a 24-hour cycle. All the data points, which were measured during a certain hour, were found, and then the median and the quantile values were calculated. The 75% quantile concentrations include, with a high probability, the data that correspond to times of higher rates of intermediate ion formation (IIF), while the 25% quantile is more likely to include data from times with no IIF. As such, although no

strict division between NPF events and non-event was made, we could derive information on the ion concentrations with respect to the probable strength of IIF.

In addition, we used the daily background ion concentrations, which were assumed to correspond to the concentrations when no, or little, IIF was taking place. These concentrations were determined as median values between 00:00 and 08:00. This time span was chosen based on a visual analysis of

145 the data.

**Table 1:** The four different size bins, which were used in the analysis. The data was measured by the Neutral cluster and Air Ion Spectrometer (NAIS) and the bins are based on the data inversion used.

| Geometric mean mobility diameter (nm) | Limits (nm) |
|---|---|
| 1.87 | $1.73 \leq d_{ion} < 2.01$ |
| 2.16 | $2.01 \leq d_{ion} < 2.32$ |
| 2.49 | $2.32 \leq d_{ion} < 2.68$ |
| 2.88 | $2.68 \leq d_{ion} < 3.10$ |



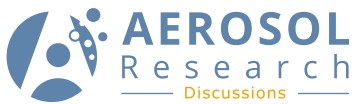

## 2.3 Horizontal ion transport

Simple linear calculations were made to illustrate how long a distance a growing ion can be transported before being measured depends on its size. We assumed a constant growth rate (GR) for the ions, and that the growing ions were transported horizontally along air masses characterized by a constant wind speed. Thus, if the initial ion size is $d_0$ and the size it is measured at is $d_1$, we can say that the farthest distance it can have traveled during its growth is

$$\text{distance} = \frac{d_1 - d_0}{\text{GR}} \times \text{windspeed}. \tag{1}$$

## 3 Results and discussion

We investigated atmospheric ion concentrations for four different diameters to determine the most suitable diameter range, for characterization of local intermediate ion formation (LIIF). Intermediate ions with mobility diameters between approximately 2 and 7 nm have in previous studies been used to capture and investigate NPF (see e.g., Kulmala et al., 2013b). In this work, we narrowed the investigated mobility diameters to between 1.7 and 3.1 nm. These limits were chosen based on our main motivations: first, as we are interested in local ion formation, we wanted the source area of the growing ions to be as small as possible. The upper limit of 3.1 nm was decided based on two assumptions: that the ions larger than 3.1 nm in diameter are likely to originate from outside the desired source area, and that the ions with smaller diameters than 3.1 nm are sensitive enough to IIF and the inclusion of larger ions is unnecessary. In addition, as mentioned before in the introductory section, we wanted to observe the ions, which were already growing to larger sizes. Previous studies have used the mobility diameter of 2 nm as the limit between small and intermediate mediate ions. Small ions tend to be present practically all the time (e.g., Hõrrak et al., 2000; Hirsikko et al., 2011) and, as such, do not guarantee the formation of larger particles associated with atmospheric NPF. However, the value of 2 nm for the limit of small ions and intermediate ions is an approximation, and thus we chose to include one size bin extending to below 2 nm in our analysis.

Our base assumption in all our analysis is that the main source of intermediate ions is intermediate ion formation (IIF). Therefore, clear and relatively sharp increases of ion concentrations (i.e., peaks) in a relatively short time period (e.g., one to three hours) are assumed to indicate elevated rates of IIF with a high probability. Other potential explanations for such features are primary sources such as traffic (Jayaratne et al., 2014), which are assumed to be negligible in Hyytiälä, and changes in meteorological conditions or the ion sink. In addition, it could be possible that the growth of the ions is stunted, and they are then transported to the measurement site from elsewhere before evaporating. While difficult to ensure, we assume that the impact of such on the statistical behavior of the ion concentrations is minor.



It should be noted that our results are mainly concerned with the statistical features of atmospheric ion concentrations made from a relatively large number of observational data. Features of atmospheric IIF on individual days, and how that is observable from ion concentrations, might differ from the statistical observations made in this study due, for example, to variations in particle formation mechanisms/pathways and meteorological conditions.

## 3.1 Diurnal cycles of ion concentrations

We investigated the statistics of diurnal cycles of ion concentrations in four different size bins between 1.7 nm and 3.1 nm. The 25%, 50% (median), and 75% quantile concentrations for the ion concentrations were determined for each 1-hour time window of a 24-hour day (see Sect. 2.2). The values based on all the data are presented in Fig. 1. Fig. 2 and 3 include the data only from March-May and September-November, respectively. The diurnal ion concentrations for December-February and June-August are presented in the supplementary material (Fig. S1 and Fig. S2, respectively).

In Fig.1, aside from the 25% quantile concentrations of $d_{bin} \approx 1.87$ nm ions, increases in concentrations during the daytime (approx. between 10:00 and 15:00) can be clearly seen. For the median concentrations, the increase is roughly 0.5 cm$^{-3}$ in all four size bins. In Fig. 2 (spring) a similar increase in concentrations during the daytime is observed, however it is more clear compared to Fig. 1. In Fig.2, the increase in the median concentrations is roughly 2 cm$^{-3}$ in all four size bins. Based on previous research, we know that NPF events often occurs around midday, and that the main source of ions in the intermediate size range is due to IIF. In addition, we know that in Hyytiälä the spring period has the most frequent NPF events. Therefore, we can safely assume that these peaks indicate that the rate of IIF is increased during, either on site or with the growing ions being transported to the site from elsewhere. The daytime peaks during autumn (Fig. 3) are weak, and completely absent in the concentrations of $d_{bin} \approx 1.87$ nm ions. This is as expected based on the fact that NPF during the autumn is less common due to the lower precursor concentrations and photogenic activity compared to spring.

In Fig. 1, we see peaks in the 75% quantile ion concentrations also during the evening (around 20:00). These peaks are stronger for the smaller size bins, while from the concentrations in $d_{bin} \approx$ 2.88 nm the peak is barely noticeable. These peaks suggest that there is potentially slightly elevated rates of IIF also in the evening, however the efficiency of growth of particles to larger diameters appears very low. The daytime peaks of the 75% concentrations in Fig. 1 show an increase by a relatively similar amount in all four size bins. However, the evening peak for $d_{bin} \approx 1.87$ nm in the negative polarity shows an increase by over 5 cm$^{-3}$, while for the concentrations in $d_{bin} \approx 2.16$ nm the increase is less than 1 cm$^{-3}$ (Fig. 1). Evening ion clustering, attributed to organic emissions, has in previous studies been observed to take place at the site (Mazon et al., 2016; Rose et al., 2018). The effect of the evening clustering is likely to have little effect on the total production of larger particles, which could affect e.g., climate.



Next, we will discuss the differences between the four investigated size bins more. From Fig. 1-3 we can see that the concentrations of smaller size ions are overall higher than for larger ions. This is the most apparent between the concentrations in $d_{bin} \approx 1.87$ nm and the concentrations in other size bins, while the differences between the concentrations from the other three size bins are much smaller. Fig. 4 shows the median hourly values of the ion concentrations divided by the daily background concentrations (see Sect. 2.2). For the ion concentrations to be good for characterizing IIF, the difference between the background and the peaks corresponding to the elevated rates of IIF should be as clear as possible. We see that for $d_{bin} \approx 1.87$ nm, the daily peak concentration is less than 1.1 times the background ion concentration. For the three larger size bins, the peak concentration is between 1.5 to 1.7 times higher than the background concentration, with the value increasing with the diameter. However, it should be noted that the background for $d_{bin} \approx 1.87$ nm ion concentrations is likely overestimated to some extent as the increased concentrations from the evening decrease slowly during the night. Regardless, it seems probable that on average it may be more difficult to detect elevated rates of IIF, especially in the case of weak IIF, from the concentrations in $d_{bin} \approx 1.87$ nm compared to the larger size bins. This is supported by Fig. 3, which shows that during autumn there is no visible daytime peaks for $d_{bin} \approx 1.87$ nm in the median and 25% quantile concentrations, unlike for the other, larger size bins.

The evening ion cluster formation is, as aforementioned, the most apparent for the concentrations in $d_{bin} \approx 1.87$ nm and mostly disappeared by $d_{bin} \approx 2.88$ nm. For the concentrations in $d_{bin} \approx 1.87$ nm, the evening peaks are equal or higher than the daytime peaks. The behavior and diurnal pattern of the ion concentrations in $d_{bin} \approx 1.87$ nm is different from the ion concentrations in the three other size bins. Therefore, the increasing ion concentration in this size bin might not necessarily indicate that there is any considerable growth of ions above 2 nm size. In addition, if we use the concentrations in size bin $d_{bin} \approx 1.87$ nm to characterize IIF, we cannot get an accurate view of the periods with the highest rates of IIF. On the contrast, based on the three larger size bins we can identify the periods with the highest rates of IIF. While the concentrations in the size bin $d_{bin} \approx 1.87$ nm would be a good choice for investigating evening ion cluster formation, we argue they are less suited for characterization of IIF. Another important implication of our results is that ions smaller than 2 nm are arguable small ions. Based on the NAIS measurements, the separation between small and intermediate ions appears to be at the mobility diameter of 2 nm, as has been used in previous studies (e.g., Leino et al., 2016).

Next, we will discuss the differences between the two polarities. From Fig. 1, the first obvious difference between the concentrations in the two polarities is that the positive ion concentrations appear to be higher compared to the negative ion concentrations. This holds true both for all four size bins and for all hours. In addition, the difference between the peak concentrations and the lower concentrations appears to be higher for negative ions compared to positive ions. If we only look at the 75% quantile concentrations during spring (Fig. 2), we can see that, aside from $d_{bin} \approx 1.87$ nm, the peak concentrations for the negative ions are equal, or even higher, than for the positive ions. This is despite the overall lower concentration of the negative ions.



Fig. 4 shows that the concentrations of the daytime peak, which we assume to indicate the occurrence of daytime IIF, are higher compared to the background concentration for the negative ions than for the positive ions. For example, for the concentrations in $d_{bin} \approx 2.88$ nm, the peak median concentration is around 1.4 times the background for positive ions versus around 1.65 times for negative ions. Fig. 5 shows the 75% quantile values divided by 25% quantile values. We have assumed that the main source of intermediate ions is IIF. Therefore, the different quantile concentrations can be used to derive insight into the differences in the concentration signals between the times of strong IIF versus little IIF. We see that for all the four size bins the difference between the 75% and the 25% concentrations is higher for the negative ions compared to the positive ions. For example, for the negative concentrations in $d_{bin} \approx 2.16$ nm, the 75% quantile concentrations are approximately 10 times higher than the 25% quantile concentrations. For the positive ion concentrations in the same size bin, the difference is only by a factor of 5.

Based on the analysis presented here, at least in Hyytiälä, the difference between the times of IIF taking place and little to no IIF can be expected to be higher for the negative ion concentrations than the positive ion concentrations. This suggests that the negative ion concentrations are better suited for characterization of IIF than the positive ion concentrations. These observations and conclusions are in agreement with previous studies, such as Hirsikko et al. (2007), where intermediate ion formation was detected slightly more often for negative than positive ions at the Hyytiälä measurement stations.

We postulate that the influence of constant background concentrations could be larger for positive ions due to their larger mobility diameters compared to negative ions (Hõrrak et al., 2000; Harrison and Aplin, 2007), extending the background to larger diameters. This would explain our observations on the differences between the positive and the negative ion concentrations, at least to some extent. In addition, the electrode effect is known to cause discrepancies in the concentrations of the ions of the two polarities near the ground surface (Israël, 1973). However, previous studies have neglected the effect inside the boreal forest environment (e.g., Tammet and Kulmala (2005); Tammet et al., 2006). It should be noted that there might be differences in how the concentrations between the polarities differ based on the measurement site. Therefore, while for Hyytiälä the negative ion concentrations appear a preferable choice for the characterization of IIF, the same might not be true at all locations. Further studies are needed.

Based on the analysis presented in this section, we argue that out of the four investigated size bins three are suited for the characterization of IIF ($d_{bin} \approx 2.16$ nm, $d_{bin} \approx 2.49$ nm, $d_{bin} \approx 2.88$ nm). In addition, for Hyytiälä the negative polarity is arguably the better choice compared to the positive polarity. However, whether this can be generalized to other environments is uncertain. In the next section the use of ion concentrations in the characterization of IIF on the local scale (LIIF) is discussed.



### 3.2 Transport of ions and the impact on ion footprint

In the previous section, we have shown that the ion concentrations in the size bins $d_{bin} \approx 2.16$ nm, $d_{bin} \approx 2.49$ nm, and $d_{bin} \approx 2.88$ nm , can be used to characterize IIF. However, the main objective of this study is to find a size, or size range, which is most suited for the characterization of local IIF (LIIF). Therefore, it is critical to consider the effect of transport on the measured ion concentrations for different diameters. In this section, transport refers solely to the horizontal transport of a growing air ion or neutral particle, which is ionized before its detection and will be referred to as an ion for simplicity. We note that the ions can also be transported in the vertical direction in the atmosphere. However, IIF related to the detailed description of three dimensional motion of air parcels is out of the scope of the present investigation.

The larger the ion is, the longer the time it has been growing. Consequently, the potential distance the ion may have traveled during its growth increases with the size of the ion. We have illustrated this point by very simple linear calculations (see Sect. 2.3) shown in Fig. 6. In the calculations, initial size of 2 nm has been assumed based on both previous studies (see e.g., Kulmala et al., 2013b) and our results from Sect. 3.1. It should be noted that Fig. 6 presents a rough estimate, and for a more accurate estimation of the transport of growing ions, other factors such as surface roughness and canopy height would need to be considered. For our purposes in this study, a rough estimate is sufficient.

If a wind speed of 3 m/s and GR of 2 nm/h, which is close to the average particle GR in Hyytiälä (Manninen et al., 2009a), are assumed, then the observed ions in the size bin $d_{bin} \approx 2.88$ nm have traveled during their growth from 2 nm to 2.88 nm a distance of approximately 5 km (Fig. 6). In the same conditions, ions in the size bin $d_{bin} \approx 2.49$ nm have been transported from a distance approximately between 1.5 to 3 km during their growth. Most of the ions in the size bin $d_{bin} \approx 2.16$ nm would have traveled 1 km, or less. However, if the wind speed is 1 m/s, most of the ions in all the investigated size bins would have likely traveled less than a 1 km distance, and most of the ions in $d_{bin} \approx 2.16$ nm could be assumed to have traveled less than 500 meters.

We have illustrated that the distances, which growing ions have traveled during their growth, are strongly dependent on the size. Therefore, even small increases in the diameter of the detected ions could mean that they have been transported hundreds of meters more during their growth. The closer the detected ions are to the size at which they have started to grow, the more probable it is that they can be attributed to LIIF. Based on this, we argue that the ions concentrations in $d_{bin} \approx 2.16$ nm, corresponding to the size range of 2.0-2.3 nm, are better suited for detecting and characterization of LIIF, as compared to the concentrations in $d_{bin} \approx 2.49$ nm or $d_{bin} \approx 2.88$ nm. The application of our results will be discussed in Sect. 4.

### 3.3 Impact of data amount on ion diurnal cycle

Based on the discussion in the previous section, the ion concentrations in $d_{bin} \approx 2.16$ nm are recommended be included for characterization of LIIF. Two matters to consider remain: first, we





need to evaluate whether it makes a difference, or not, to use ion concentrations only from $d_{bin} \approx$ 2.16 nm, versus including data also from the larger bins in the analysis. Second, using data only from one size bin could potentially increase the influence of statistical noise, especially if data are sparse, and thus lead to higher uncertainties in the observations of IIF.

Fig. 7 shows the median diurnal curves for the negative ion concentrations in $d_{bin} \approx$ 2.16 nm, and between the diameters 2.01–2.50 nm and 2.05–2.68 nm. The latter are based on the nearest neighbor interpolation and take into account the concentrations in both $d_{bin} \approx$ 2.16 nm and $d_{bin} \approx$ 2.49 nm. Both curves with all data, 50%, 10% and 1% of it are included. The 50%, 10%, and 1% samples of the full data were based on a random sampling of data from all data points.

Fig. 7 shows that including the concentrations only from $d_{bin} \approx$ 2.16 nm, or from both $d_{bin} \approx$ 2.16 nm and $d_{bin} \approx$ 2.49 nm has a minor effect on the averaged behavior of the negative ion concentration. As such, if we use only the concentrations from $d_{bin} \approx$ 2.16 nm versus from e.g., both $d_{bin} \approx$ 2.16 nm and $d_{bin} \approx$ 2.49 nm, there should statistically be no major effect on the observed behavior of ion concentrations during IIF. In addition, Fig. 7 shows that reducing the amount of data does not seem
to result in a more considerable amount of noise if only data from one size bin is used compared to if data from two size bins is used. Thus, we argue that using the ion concentrations in $d_{bin} \approx$ 2.16 nm, which corresponds to a diameter range of 2.01–2.32 nm, are the best choice for characterization of LIIF.

## 4 Atmospheric relevance and applicability

In this section, we will discuss a couple of important issues that need to be considered in application of our results. In addition, example cases for the application of the results will be discussed.

First, one should consider, which polarity data to use. The answer to this question, assuming that data for both negative and positive polarity is available, depends on a couple of factors. First, the application should be considered. Sometimes, using data for both polarities is desirable. Differences
in the polarities could, for example, be used to derive insight into the growth mechanisms during LIIF or to study the effect of polarity on LIIF. However, other times, use of data for only one polarity is preferable, which could for example be the case due to the desire for a more straightforward application and analysis. Our results have shown that in Hyytiälä, Finland, the negative ion concentrations are likely more sensitive to elevated rates of IIF compared to positive
ion concentrations, and therefore are the better choice for characterizing LIIF. Whether this is true also at other sites is recommended to be verified through statistical analysis and comparison before application.

In addition to IIF, intermediate ion concentrations could be affected by snowstorms and rain, and therefore these should be filtered out from the data either by use of additional data or visual analysis
of the ion concentrations.

The potential source area (i.e., the area from which the ion could have been transported from during its growth) of the 2.0-2.3 nm ions should be considered. While we consider this size range to be the best option for characterization of LIIF, these ions could also have been transported during their



growth. The distances are just smaller than for the larger ions. Therefore, it is recommended to
consider the features, such as the landscape, vegetation, and primary emission sources (e.g., traffic)
of the potential source area of the detected ions, and how they impact the observed concentrations
of the 2.0-2.3 nm ions.

In addition, it should be kept in mind that we have defined LIIF so that the initial phase of the
growth of the ion from a cluster to larger sizes occurs within a proximity to the measurement site.
As such, local does not strictly mean that the observed IIF would be free from influences of air
outside the source area of interest. Air masses from outside the source area transport larger pre-
existing particles and precursor chemical compounds, influencing both the rate at which the
growing clusters are coagulating with larger particles and the rate that they are growing to larger
sizes. For example, in Hyytiälä, air masses arriving from northwest direction have been shown to
favor NPF due to these air masses having a low surface area of pre-existing particles (Dal Maso et
al., 2007; Dada et al., 2017). In addition to precursor compounds emitted within the area,
transported precursor compounds could also affect the number of new clusters. Therefore, one
should not interpret the 2.0-2.3 nm ion concentrations or LIIF as independent of influences from
outside the assumed source area.

Next, two example cases for the application of our results are discussed. First, if we want to
estimate the contribution to total regional particle production by different environments, such as a
boreal forest and a wetland, the 2-2.3 nm ion concentrations can be used to represent the particle
production if some assumptions are made (see Kulmala et al., 2024). If the average ion sink and the
ion growth rate are similar in these environments, the 2.0-2.3 nm ion concentrations should be
proportional to the particle production. This information can then for example be used to estimate
the contribution of different environments to e.g., CCN production or aerosol radiative forcing.

In addition, the growth of the 2.0-2.3 nm ions from clusters occurs mainly within an area, which is
similar in size to the footprint area of tower-based eddy covariance measurements. Therefore, the
ion concentrations and the eddy covariance fluxes can be assumed to be mostly under the influence
of the same environmental conditions. Kulmala et al. (2020) recently developed a concept of
CarbonSink+, which accounts for multiple boreal forest climate-biosphere feedbacks, including
atmospheric particles and NPF. Our results can therefore be applied to represent particle production
within a similar area as $CO_2$ flux, and other fluxes, to study their combined climate impacts (see
Kulmala et al., 2024).

## 410    5 Conclusions

Our main objective in this study was to evaluate the suitability of ion concentrations of different
sizes for characterization of local intermediate ion formation (LIIF). We studied the ion
concentrations in four small size ranges between the mobility diameters 1.7 nm and 3.1 nm. Ion
number size distribution data measured by a Neutral cluster and Air Ion Spectrometer (NAIS) at the
SMEAR II measurement station in Hyytiälä, southern Finland, was used.



We found that ion concentrations in the size ranges of 2.0-2.3, 2.3-2.7, and 2.7-3.1 nm can be used in finding periods with elevated rates of intermediate ion formation (IIF), and to evaluate the potential strength of IIF. Ions below 2 nm were found less suitable for such purposes. Ions below 2 nm have higher background concentrations, and appear less affected by IIF compared to larger ions. In addition, the dynamics of sub-2 nm ions are different from larger ions. These observations indicate that 2 nm is the size, which separates small ions and intermediate ions. Compared with positive ions, negative ions were found to be more sensitive to IIF at the SMEAR II measurement station, however whether this is also true at other locations remains to be verified. The impact of transport on concentrations of ions was discussed. The potential distance that the detected ions could have been transported by air masses during their growth gets the longer the larger the ions are. Therefore, we argued that the ions in the size range of 2.0-2.3 nm are the best option for characterization of LIIF, and thus also for studying local NPF.

## Data availability

The ion number concentrations used in this study are available at https://doi.org/10.5281/zenodo.8059335 (Tuovinen et al., 2023).

## Author contributions

ST conducted the data analysis and wrote the paper. JL was responsible for the ion measurements. VMK and MK designed the study. All authors contributed to discussion of the results and provided input for the paper.

## Competing interests

At least one of the (co-)authors is a member of the editorial board of Aerosol Research. Authors have no other competing interests to declare.

## Acknowledgments

We acknowledge the following projects: ACCC Flagship funded by the Academy of Finland grant number 337549 (UH) and 337552 (FMI), Academy professorship funded by the Academy of Finland (grant no. 302958), Academy of Finland projects no. 1325656, 311932, 334792, 316114, 325647, 325681, 347782, "Quantifying carbon sink, CarbonSink+ and their interaction with air quality" INAR project funded by Jane and Aatos Erkko Foundation, "Gigacity" project funded by Wihuri foundation, European Research Council (ERC) project ATM-GTP Contract No. 742206, and European Union via Non-CO2 Forcers and their Climate, Weather, Air Quality and Health Impacts (FOCI). University of Helsinki support via ACTRIS-HY is acknowledged. University of Helsinki Doctoral Programme in Atmospheric Sciences is acknowledged. Support of the technical and scientific staff in Hyytiälä are acknowledged.



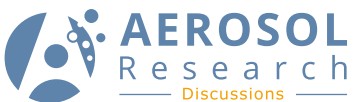

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





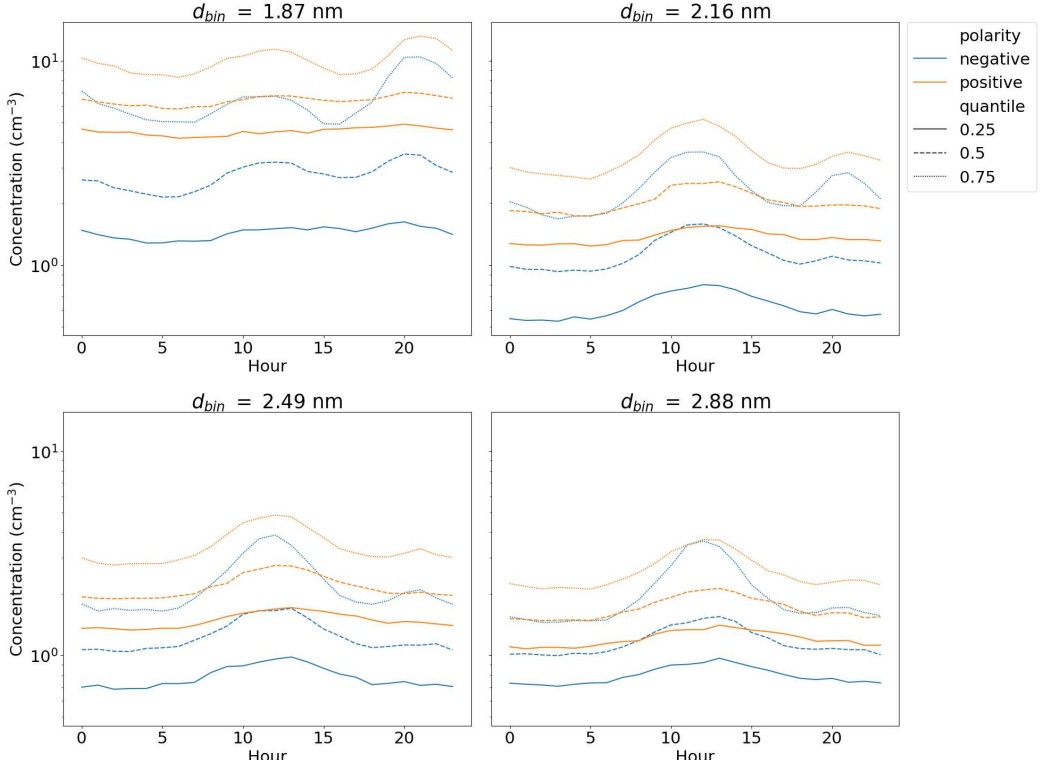

**Fig. 1:** Hourly ion concentrations in four size bins with geometric mean mobility diameter $d_{bin}$ based on median, 25%, and 75% quantiles. The ion concentrations were measured by Neutral cluster and Air Ion Spectrometer (NAIS) at SMEAR II measurement station in Hyytiälä, Finland from 2016 to 2020. Data from all seasons is included and no distinction between the days that were classified as NPF events days, or not, was made.



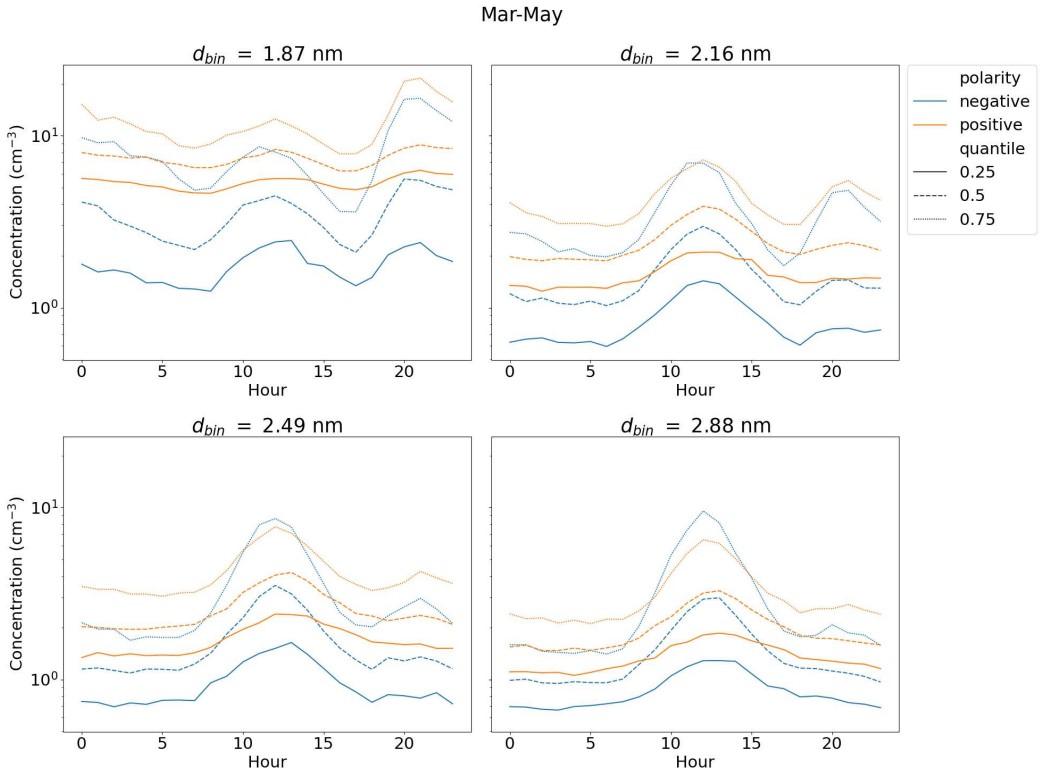

**Fig. 2:** Hourly ion concentrations from March-May in size bins with geometric mean mobility mobility diameter $d_{bin}$ based on median, 25%, and 75% quantiles. The ion concentrations were measured by Neutral cluster and Air Ion Spectrometer (NAIS) at SMEAR II measurement station in Hyytiälä, Finland from 2016 to 2020.



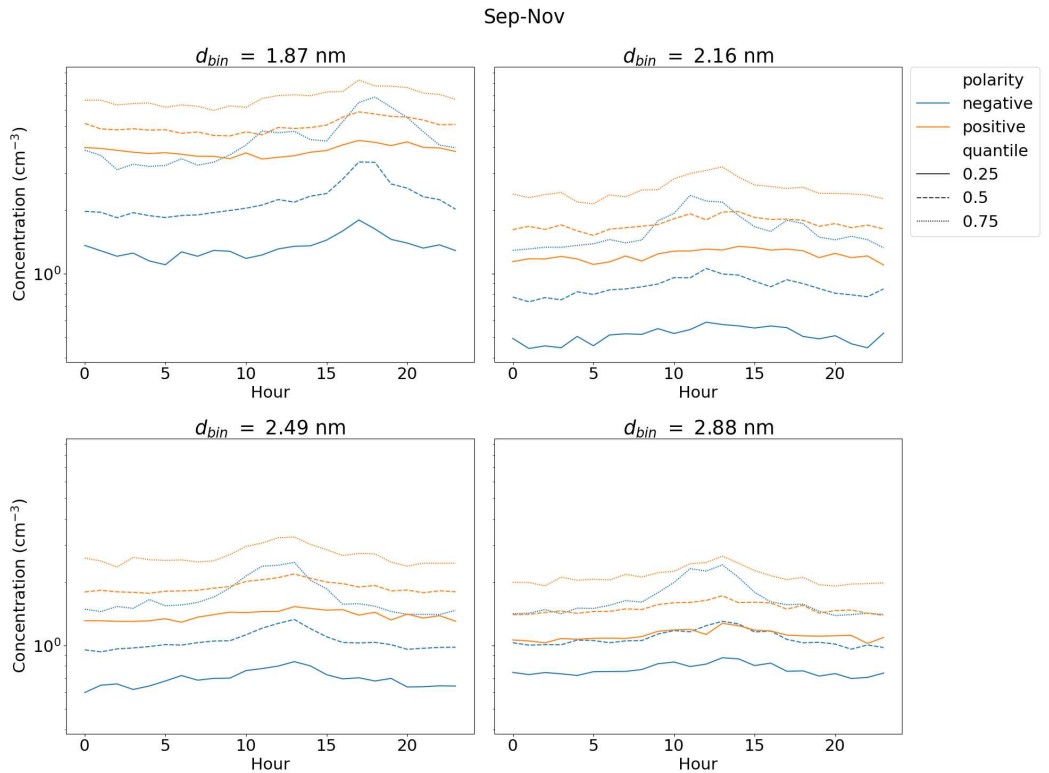

**Fig. 3:** Hourly ion concentrations from September to November in size bins with geometric mean mobility diameter $d_{bin}$ based on median, 25%, and 75% quantiles. The ion concentrations were measured by Neutral cluster and Air Ion Spectrometer (NAIS) at SMEAR II measurement station in Hyytiälä, Finland from 2016 to 2020.

480



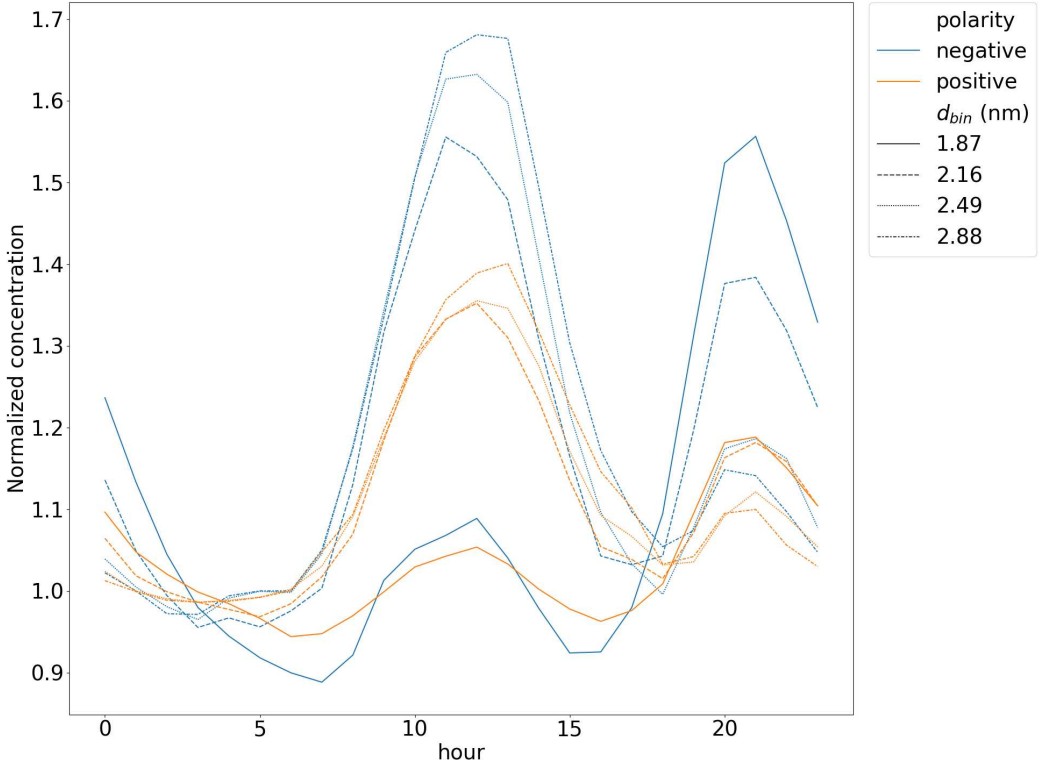

**Fig. 4:** The median hourly ion concentrations normalized by the background ion concentration. The geometric mean mobility diameters of the different size bins are denoted with $d_{bin}$. The ion concentrations were measured by Neutral cluster and Air Ion Spectrometer (NAIS) at SMEAR II measurement station in Hyytiälä, Finland from 2016 to 2020.





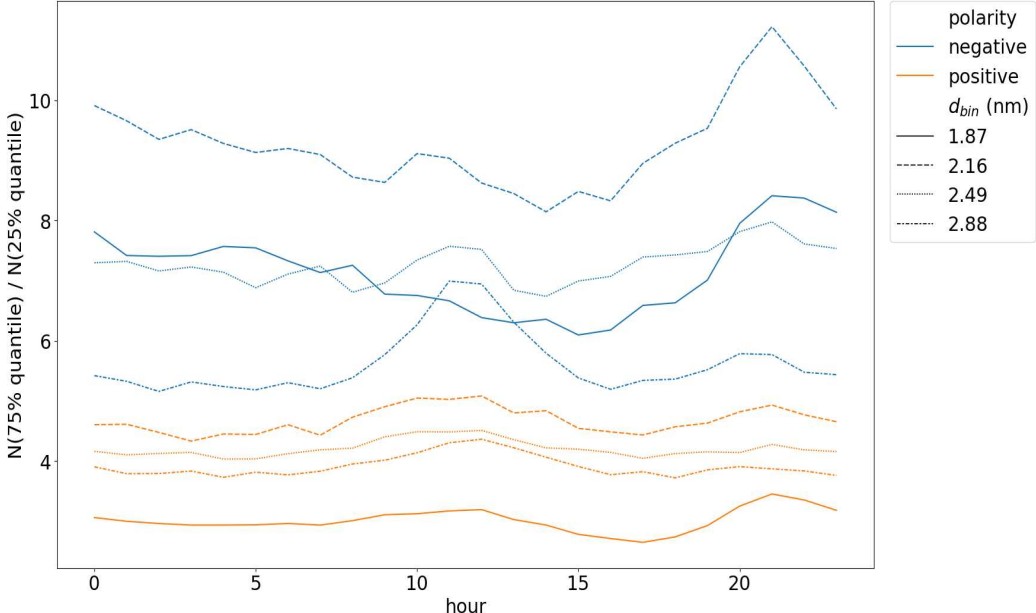

**Fig. 5:** Hourly 75% quantile ion concentrations divided by the 25% quantile concentrations of the same hour. The geometric mean mobility diameters of the different size bins are denoted with $d_{bin}$. The ion concentrations were measured by Neutral cluster and Air Ion Spectrometer (NAIS) at SMEAR II measurement station in Hyytiälä, Finland from 2016 to 2020.



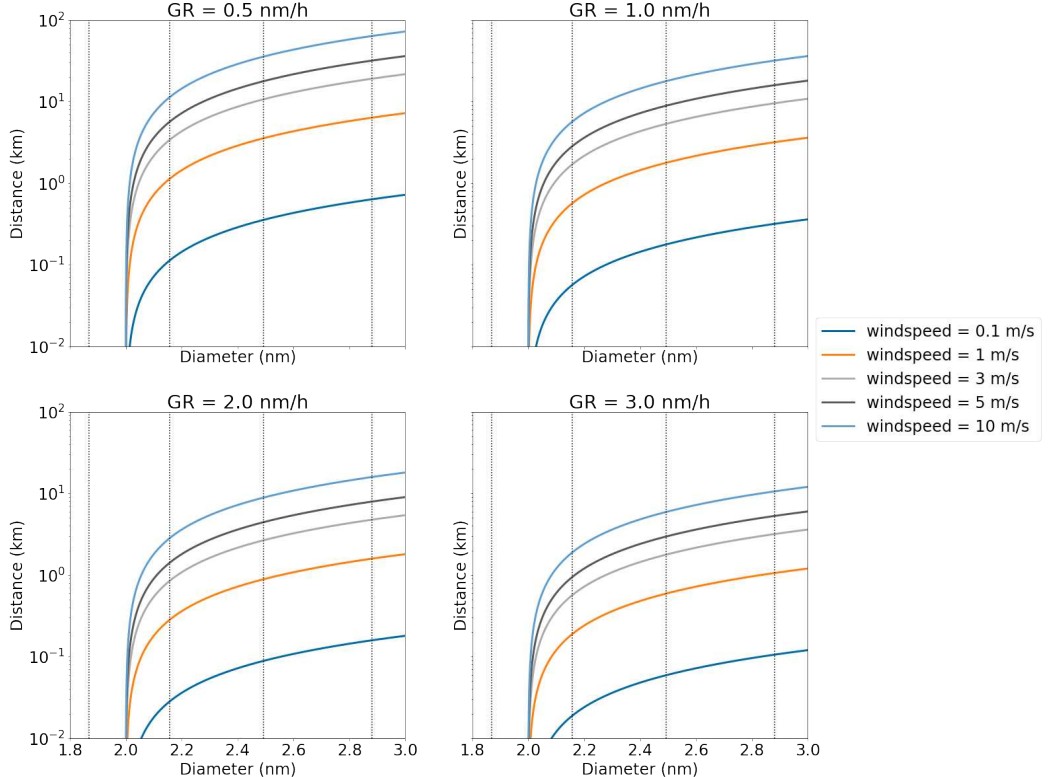

**Fig. 6:** The distance a growing atmospheric ion or a neutral particle can be transported by horizontal winds assuming initial mobility diameter of 2 nm. Growth rate of the ion/particle is denoted by GR and it is assumed to stay constant with increasing size. The vertical lines mark the geometric mean mobility diameters of the four size bins of NAIS data, which were use in the study.

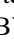




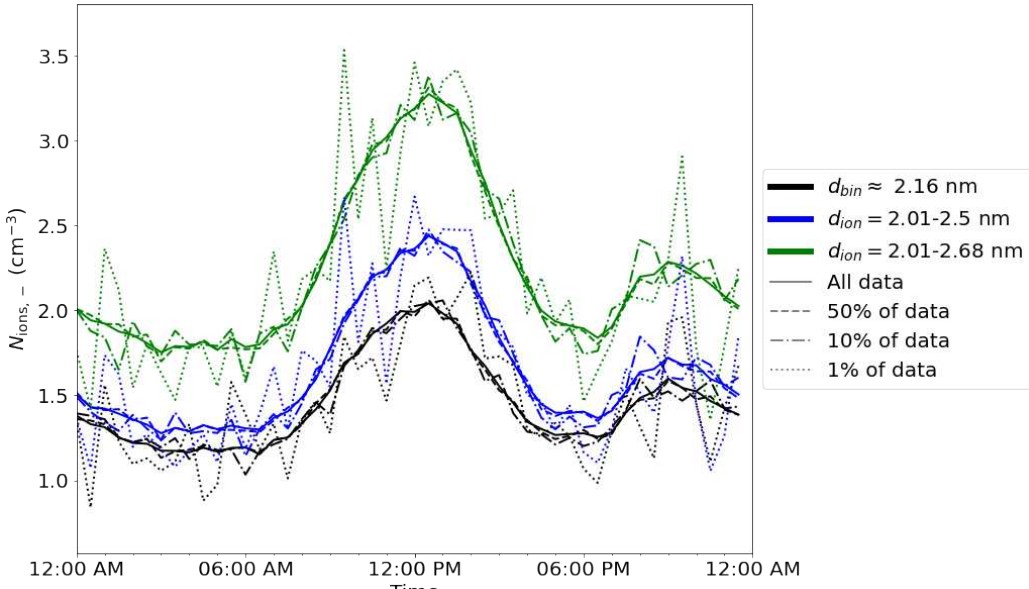

**Fig. 7:** Median daily cycle of concentrations of negative ions in size bin with geometric mean diameter $d_{bin} \approx 2.16$ nm and between size limits 2.01-2.50 nm and 2.01-2.68 nm, which include data from both the size bin $d_{bin} \approx 2.16$ nm and the size bin $d_{bin} \approx 2.49$ nm. Data is from 2016 to 2020, and it was measured with Neutral cluster and Air Ion Spectrometer (NAIS).