# Peer review of "Intermediate ions as indicator for local new particle formation"

_Aerosol Research, 2024_

## Author Comment (AC1)

**Response to referee comments**

We thank both referees for their constructive comments. They have majorly helped us to improve on our manuscript and to address its weak points. Please find the detailed answers to the comments below. Below, referee comments are in black, author comments are in blue, while direct quotes from the manuscript are marked with *italics.*

Santeri Tuovinen, on behalf of all the co-authors,

**Referee #1**

**General comments**

Touvinen et al use data from a year of field measurements to describe the development of intermediate size ions during new particle formation events. The goal is to determine the optimal size of ions to use for the characterization of local intermediate ion formation (LIIF). The data is well presented, and the paper is generally well written and easy to follow. As mentioned in the paper, new particle formation is an important topic and is still not fully understood, so new ways of looking at these kinds of events are welcome.

I am missing a discussion of why a characterization of LIIF is a useful tool for describing new particle formation events, as compared to looking at neutral particles in the same size range. Instrumentation is probably one good argument as it is typically easier to measure small/intermediate ions than neutral particles in that size range. And details of ions can also reveal information about the role of ions in the formation of the particles.

The main result is that ions of d~2.16 nm are best suited for studying local intermediate ion formation. It is perhaps not surprising that the smallest size range where a significant signal is seen is optimal, considering the "local" boundary condition, as made clear by Eq. 1 of the paper. Could it be phrased slightly more general, something like that the optimal size range is "the smallest size bin above the upper size range of small air ions"? And will the cut-off between small and intermediate ions always be around 2 nm or how chemistry dependent is that?

Overall, I think this is a fine contribution to the description of the behavior of ions during new particle formation events that can be published after a few elaborations and corrections.

**Specific comments and technical corrections/suggestions**

*L14*: --> "measured by the Neutral cluster and Air Ion Spectrometer" Thank you for noticing this. It has been fixed.

*L17*: "Local" is mentioned several times and here described as "close proximity to the measurement site". Is there a more technical way to describe what is meant by "local"?

Thank you for bringing this issue to our attention. In the revised manuscript, the definition of local has been refined.

(L18-L19): *in a close proximity (e.g., within 500 m to 1 km) to the measurement site, i.e., locally.*

(L90-L91): *In the context of our study, local means within a close proximity (in practice, within 500 m to 1 km) as well as within the same environment.*

The definition has been further explained (L111-L114): *Ideally, the distance ions have been transported should be as small as possible, however as aforementioned, requiring a smaller maximum distance than 500 to 1 km is not practical considering the timescales of particle growth and air mass transport.*

*L33*: Specify the fraction.

The estimated fraction has now been specified (L33-L34): *A large fraction, estimated to be over a half, of the global aerosol number concentration is due to atmospheric new particle formation (NPF)*

*L50*: --> "contribute with most of the particle production"?

The sentence has now been fixed (L51) : *… contribute most to the particle production ...*

*L89*: Charged ions --> "charged particles" or simply "ions". Thank you for noticing this, it has now been fixed.

*L90*: Some neutral particle formation is bound to happen at the same time, at least from neutralization of charged particles.

Thank you fot this comment. It is true. We have added some discussion on the relationship between (L)IIF and neutral particle formation to the revised manuscript.

(L98-L105): *Usually atmospheric NPF is dominated by neutral pathways (Kulmala et al., 2013), and as some of the neutral particles are charged, simultaneous IIF can be observed. However, even if the ion-induced pathways dominate, collisions between oppositely charged ions, neutralizing the ions, are bound to take place, and result in the formation of neutral particles concurrently to IIF. Therefore, LIIF can be used to identify local NPF, regardless of the nucleation pathway. However, the total particle production rate cannot be directly derived from the observed intensity of LIIF, unless the particles are at equilibrium charge fraction (Kerminen et al., 2007; Leppä et al., 2013), which is usually not the case (Leppä et al., 2013).*

*L148-149*: This line is confusing to read, I think it is supposed to say that "Simple linear calculations were made to illustrate the size dependency on how far a growing ion can be transported before being measured" or something like that?

Thank you fot suggestion. This sentence has now been reworded, and it now reads (L190-L191): *Simple linear calculations were made to illustrate the size dependency of how far a growing ion can be transported before being measured.*

*L162*: The reasoning for the size limits chosen probably belongs in the Methods section, at least I was wondering about it since line 131. The upper limit of 3.1 nm could be explained more. What is the "desired source area" (L164)? I suppose the 3.1 nm are found based on your eq. 1 using typical growth rates and windspeeds from the area, but these should be listed so the reader can better understand the reasoning behind the 3.1 nm. From the data in figs 1-4 the upper limit seems reasonable.

Thank you for your suggestion. We have now moved the discussion on the choice of size limits under a new Section 2.3 Choosing the investigated diameters. In addition, we have noted in the revised manuscript (L180-L181): *that in the initial phases of this study, ions with diameters up to 4*

*nm, which was the upper limit used by e.g., Dada et al. (2018), were considered, and later excluded.*

*L253*: What is the reason behind the larger positive ion concentration? Typically, small positive ions are larger than small negative ions (as you note in L284) so a shift in the spectrum between the two polarities could be an explanation. This should be possible to check from the NAIS data, I think.

We believe that a shift in the spectrum is the most likely reason for larger positive ion concentrations. We have added some further discussion on this, and added a new figure to supplementary information of the manuscript.

(L316-L324): *We postulate that the influence of constant background concentrations could be larger for positive ions due to their larger mobility diameters compared to negative ions (Hõrrak et al., 2000; Harrison and Aplin, 2007), extending the background to larger diameters. This is supported by Fig. S3, showing the median hourly concentrations of both polarities for diameters 0.8-1.2 nm, 1.2-1.6 nm, and 1.6-2 nm.We can see that the concentration of the smallest ions is higher for negative ions, whereas the positive ion concentration is higher than the negative one for both 1.2-1.6 nm and 1.6-2 nm ions. This suggests a shift in the small ion spectrum for positive polarity compared to negative polarity, and would explain our observations on the differences between the positive and the negative ion concentrations, at least to some extent.*

[Figure]

**Fig. S3:** Median hourly concentrations of positive and negative small ions for diameter ranges 0.8-1.2 nm, 1.2-1.6 nm, and 1.6-2 nm. The ion concentrations were measured by a Neutral cluster and Air Ion Spectrometer (NAIS) at the SMEAR II measurement station in Hyytiälä, Finland from 2016 to 2020.

*L298*: That the signal in negative ions is stronger than in the positive ions can be a sign that the nucleation is due to ion-induced nucleation (as described by some of the authors in Kerminen et al,

JGR 112, D21205, 2007 and measured in Enghoff and Svensmark, J. Aerosol Sci 114 p13-20, 2017) which typically has a negative preference. This would in turn suggest that the negative polarity would be the better choice in most cases (with the exception e.g. of the cases mentioned in the beginning of Sect. 4).

Thank you for bringing this up. We have now added brief discussion on this. (L329-L333): *Previous studies (e.g., Enghoff and Svensmark, 2007) have shown that ion-induced nucleation can result in a higher overcharge (i.e., higher concentration compared to equilibrium) of negative ions compared to positive ions. This suggests that negative ions might be more sensitive to IIF in general, not just in Hyytiälä as shown in this study, at least if ion-induced nucleation is a major contributor to IIF.*

*L402*: Should probably start with "Second" instead of "In addition" to make it clear that this is the second of the cases introduce din L395. We thank you for bringing this up, and it has now been changed.

*Figs 1-4*: Thicker lines would be appreciated; it is hard to tell the difference between the types of dotted/dashed lines when printed. Thicker lines have now been implemented.

**Citation**: https://doi.org/10.5194/ar-2024-4-RC1

**Anonymous Referee #2**

In "Intermediate ions as indicator for local new particle formation", Santeri Tuovinen and co-authors argue that ambient ions with mobility diameters just above 2 nm are most suitable as indicators of new particle formation initiated locally (within kilometers). Their analysis is based on several years worth of data at a boreal forest site in southern Finland. Concentrations of smaller ions tended to respond to processes not related to sustained growth, whereas larger ions would be more susceptible to sources and processes farther afield.

The study has thus limited scope, but it presents a new perspective for ambient ion mobility size spectrometry data. And although only based on measurements at a single site, it is a prominent site in the field of atmospheric sciences and has ample statistics available. With that, I believe the study will be of interest to the expected readership of "Aerosol Research".

Overall, it is not too badly written either.

My main concern evolves around the often repeated formulation "characterization of LIIF" (and variations thereof). Pivoting around those tend to be unclear lines of reasoning -- increasingly so the farther the analysis proceeds. Similarly, I find the "locality" aspect insufficiently defined.

The analysis overall and general conclusions (if I understand them correctly) are interesting and worth reporting, but they are also really quite straightforward, given the convenient (yet likely justified) assumptions made. In light of that, the paper is considerably more elaborate than it needs to be, leading to some confusion to this reviewer (and increasing irritation) with increasingly more frequent vague formulations as the paper carries on.

**My major comments** are all related to that main concern:

The Abstract is generally concise yet informative. However it makes three uses of that rather vague term "characterizing LIIF" ... What is meant? ("identifying"?)

*We thank you for bringing this up. Characterization was a poor choice of words and has now been replaced with more precise wording (L16):* ... identifying and evaluating the intensity of local intermediate ion formation (LIIF).

*All later instances of the word characterizing has been similarly reworded in the revised manuscript.*

*To further clarify the issue, we have also added the following to the beginning of the Results section (L211-L213):* ... IIF can be identified from elevated intermediate ion concentrations. Since we have assumed that IIF is the main source of intermediate ions, higher intermediate ion concentrations can be assumed to correspond to more intense IIF.

The Introduction is well structured, but there is some lack of clarity in the goals of the study. In particular at line 100 (L100), plus the rest of paragraph, I wonder again, what is meant by "characterizing LIIF"?  Differentiating LIIF from IIF? The latter being affected by "transported ions" (related to my minor comment on L96+, see below), whereas the former is not?

*We thank you for this comment. Our objectives have now been slightly reworded for clarity (L120-L128):* In this study, we will investigate intermediate ion concentrations measured in Hyytiälä, Finland, using a Neutral cluster and Air Ion Spectrometer (NAIS) (Mirme and Mirme, 2013; Manninen et al., 2016). Our aim is to find out the optimal size range of intermediate ions to be used in identifying and evaluating the intensity of LIIF. In addition, both ion polarities will be compared, and the potential impact of polarity on intermediate ion concentrations, and therefore on the sensitivity to, and the characteristics of LIIF, will be evaluated. The potential contribution of transport on the ion concentrations will be discussed. Finally, a recommendation for the best ion diameter to use in the identification and evaluating the intensity of LIIF with minimal influence from transportation is given.

*In addition, other parts of the introduction have been worked on to add clarity to the goals of this study.*

*The following paragraph has been slightly rephrased to clarify the definitions used in this study and our aims (L86-L96):* In this work, we will investigate the use of atmospheric intermediate ion concentrations for studying local NPF. There are two important issues connected to this. First, we want to exclude primary ions as well as the ions that have not (yet) been activated for growth and might not contribute to the local particle production.  Therefore, we want to observe only the ions attributable to NPF as per our definition. Second, the activation of clusters for growth should occur as locally as possible. In the context of our study, local means within a close proximity (in practice, within 500 m to 1 km) as well as within the same environment. We will refer to the formation of intermediate ions as IIF (intermediate ion formation), and to the IIF, which occurs locally, as LIIF, The separate term for intermediate ion formation compared to NPF is used to make it clear that we are observing and studying the formation of charged particles. At which intensity the formation of neutral particles is taking place at the same time, is not known for certain.

*And the two paragraphs following the above one have been mostly rewritten to further clarify the definitions of IIF and LIIF:*

*Usually atmospheric NPF is dominated by neutral pathways (Kulmala et al., 2013), and as some of the neutral particles are charged, simultaneous IIF can be observed. However, even if the ion-induced pathways dominate, collisions between oppositely charged ions, neutralizing the ions, are bound to take place, and result in the formation of neutral particles concurrently to IIF. Therefore,*

*LIIF can be used to identify local NPF, regardless of the nucleation pathway. However, the total particle production rate cannot be directly derived from the observed intensity of LIIF, unless the particles are at equilibrium charge fraction (Kerminen et al., 2007; Leppä et al., 2013), which is usually not the case (Leppä et al., 2013).*

*The intermediate ion concentrations are affected by transportation, which means that growing ions and neutral particles, which are ionized before detection, have been transported by moving air masses. Therefore, depending on the distance the ions have been transported, the factors which have lead to the activation of the clusters for growth or impacted their growth rate, might differ. Our aim is to use ion concentrations to identify LIIF. Thus, we want to minimize the impact of transportation on the observed intermediate ion concentrations. Ideally, the distance ions have been transported should be as small as possible, however as aforementioned, requiring a smaller maximum distance than 500 to 1 km is not practical considering the timescales of particle growth and air mass transport. The wider the size range of ions is, the wider their potential source area will be. Therefore, narrow ion diameter ranges should have less variation in the potential source area compared to wider ranges. We note that while transport of ions can be both horizontal and vertical, in this study our focus is on the horizontal transport.*

Results and discussion:

L157: "characterization of LIIF" again

This sentence has now been modified (L198-L200): *We investigated atmospheric ion concentrations for four different diameters to determine the most suitable diameter range for identification and evaluating the intensity of local intermediate ion formation (LIIF).*

L248, 249: "investigating evening cluster formation" and "characterization of IIF" ... As above, what do the authors mean by these? Identifying periods? Measuring rates of formation?

We have now reworded this sentence to be more exact in its wording (L279-L282): *While the concentrations in the size bin $d_{bin} \approx 1.87$ nm would be a good choice for detecting and evaluating the potential intensity of evening ion cluster formation, we argue they are less suited for detecting or evaluating the intensity of IIF.*

I also get the feeling here that we are just restating definitions. (The remainder of the paragraph further strengthens that feeling.) In this case <2nm meaning clusters, >2nm meaning II. The relation of either to onward growth (no for clusters, yes for II) requires looking at the concentration evolutions of multiple size bins. (As the authors are doing in this study, including references to previous works that made use of size distributions.) But why is it beneficial to examine, in which size bin on its own would concentration increases most likely relate to that onward growth? That is something that could be explained when motivating the study. Hence, this comment again relates to the hazy initial objectives.

Thank you for bringing this up. Single size bins from the data were investigated as they are as narrow as diameter ranges could be, and thus the variation in the source area of ions within the size ranges would ideally be as small possible. We hope to clarify this issue with the following additions to the revised manuscript (L115-L116): *The wider the size range of ions is, the wider their potential source area will be. Therefore, narrow ion diameter ranges should have less variation in the potential source area compared to wider ranges.*

*(L183) : Narrow size bin wide diameter ranges were investigated to minimize the variation in the potential source area of the growing ions.*

L304: another example of those vague formulations: "used to characterize IIF" ... Would "indicative of IIF that relates to continuous growth traditionally associated with NPF events" work, for example?

Thank you for the comment and the suggestion. This has been reworded (L385-L386): *Based on the discussion in the previous section, the ion concentrations in $d_{bin} \approx 2.16$ nm are recommended to be used for identifying and evaluating the intensity of of LIIF.*

L332-333 (and the remainder of the paragraph): To judge how "probable it is that [ions] can be attributed to LIIF", one would need to define LIIF. However, that has not been done. If I understand correctly, the goal is merely to associate ions with a start of growth as close ("local") as possible. Which of course, at least in the spirit of the transport analysis in this paper/section, is trivially the case for the smallest ions. The paragraph thus appears as an attempt to oversell that trivial circumstance.

Thank you for this comment. We have clarified the definition of local (L90-L91): *In the context of our study, local means within a close proximity (in practice, within 500 m to 1 km) as well as within the same environment.*

This has been expanded on (L112-L115): *Ideally, the distance ions have been transported should be as small as possible, however as aforementioned, requiring a smaller maximum distance than 500 to 1 km is not practical considering the timescales of particle growth and air mass transport.*

We have also rephrased the definition of LIIF (L92-L93): *We will refer to the formation of intermediate ions as IIF (intermediate ion formation), and to the IIF, which occurs locally, as LIIF,*

To Sect. 3.2 we have added (L350-L351): *As previously defined, LIIF in this study refers to IIF, where the activation of the ions for growth has occurred within maximum 500 m to 1 km.*

Additionally, IIF and the goals of this study have been expanded on (see answer to the 1. comment).

With these more precise definitions, the discussion should appear less trivial.

The Section "Atmospheric relevance and applicability" is quite useful, as it raises awareness of possible spatial heterogeneities in processes that could affect ion size distributions. But similar instances of unclarity are encountered.

L378: I do not understand the statement "ions could also have been transported during their growth". Is not the whole premise of assessing the locality of IIF based on the transport of these ions during their growth (Section 3.2)? Please clarify. (The latter part of the paragraph makes me believe the suggestion is to consider effects of heterogeneities and possible direct emission sources on apparent growth rates?)

Thank you for identifying this issue. We have now removed this vague and perhaps unnecessary statement from this paragraph and reworded it (L418-L424): *The potential source area (i.e., the area from which the ion could have been transported from during its growth) of the 2.0-2.3 nm ions should be considered. It is recommended to consider the variation in features, such as the landscape, vegetation, and primary emission sources (e.g., traffic) within the potential source area*

*of the detected ions, and how it can impact the observed concentrations of the 2.0-2.3 nm ions. For example, direct emissions of small ions might increase the observed concentrations while the GR of ions can vary based on heterogeneties in the precursor vapor, such as low volatile organic compounds, concentrations.*

L407: "represent" is unclear. "estimate"?

*We have reworded the sentence using estimate instead of represent.*

The "Conclusions" are pleasantly concise, yet rather a "summary". The first sentence emphasizes my concern about the poorly posed objective.

*We have reworded parts of the conclusions*

*(L451-L452): Our main objective in this study was to evaluate the suitability of ion concentrations of different sizes for identifying and evaluating the intensity of local intermediate ion formation (LIIF)*

*(L466-L467): Therefore, we argued that the ions in the size range of 2.0-2.3 nm are the best option for identifying and evaluating the intensity of LIIF associated with NPF.*

**Minor comments:**

Abstract:

Could add reason for why negative ions better suited (~L22).

*This has now been addressed in the abstract (L22-L24): In addition, in Hyytiälä, the negative ion concentrations are more sensitive to IIF than the positive ion concentrations due to the higher difference in concentrations between periods of IIF and the background*

Introduction:

L38: growth to which sizes is considered required for "NPF"?

*Thank you for the comment. We have now specified this (L39-L40): We consider the growth of the particles to above roughly 2-3 nm as a necessary prerequisite for NPF.*

L44: What are "quiet NPF" and "traditional NPF event times"? Should be elaborated on or the sentence skipped.

*Thank you for bringing this up. This sentence has now been reworded for clarity (L45-L46): In addition, there is so-called quiet NPF, taking place on days typically classified as NPF non-event days (Kulmala et al., 2022).*

L96 + paragraph around it: I find this part difficult to follow. Not sure I understand correctly what are considered "transported ions" vs. non-transported ions. Or there might be a mistake here. Please double-check, and ideally clarify.

*Thank you for bringing this to our attention. This paragraph has now been rewritten (L107-L115): The intermediate ion concentrations are affected by transportation, which means that growing ions and neutral particles, which are ionized before detection, have been transported by moving air*

*masses. Therefore, depending on the distance the ions have been transported, the factors which have led to the activation of the clusters for growth or impacted their growth rate, might differ. Our aim is to use ion concentrations to identify LIIF. Thus, we want to minimize the impact of transportation on the observed intermediate ion concentrations. Ideally, the distance ions have been transported should be as small as possible, however as aforementioned, requiring a smaller maximum distance than 500 to 1 km is not practical considering the timescales of particle growth and air mass transport.*

Methods:

2.1, 2nd paragraph: could mention that the NAIS measured ions of both polarities simultaneously.

Thank you. It is now mentioned.

L128: Does that mean that new classification methods differ from old methods, and that old method was used here to define "NPF day"? If so, is there a reference for that specific classification method? That should be clarified if the classification or definition of "NPF day" will matter for this paper.

All days have been used, regardless of if any classification method would have classified the day as NPF event day, or not. This paragraph in question has now been reordered for clarity (L149-L151): *Recent advances have shown that NPF does occur even during the days classified as non-event days (Kulmala et al., 2022). Therefore, the data were used from all the available days, and no distinction was made based on whether the days had been classified as NPF days or not.*

2.2, last paragraph: presumes/implies it is known when IIF takes (not) place from visual inspections ... What did that "visual analysis" consist of? Why was the statistical analysis performed if visual analysis was sufficient for identifying IIF? Or was it so that the "visual analysis" was performed on the statistical results? Also, if so, that "visual analysis" needs to be elaborated on.

Thank you for the comment. This has been clarified (L165-L167): *This time span was chosen based on a visual inspection of the statistical behavior of the ion concentrations. Time periods, when the variation in the concentrations was relatively low, were assumed to correspond to times with (statistically) little IIF.*

Results and discussion:

L180: Could be added that the coinciding of statistically observed IIF with the times of NPF during "event days" will retroactively vindicate that assumption. (Or if later text states so, could refer to that later text here.)

It has been now mentioned (L219): *This assumption is vindicated by the observations of elevated ion concentrations statistically coinciding with time periods of elevated intensity of NPF (see Sect*

L245: "cannot get an accurate view of the periods" is vague and confusing. Replace with clear and specific wording.

Thank you for bringing this issue up. In the revised manuscript, it is now written as (L285): *In addition, if we use the concentrations in size bin $d_{bin} \approx 1.87$ nm to evaluate the intensity of IIF, we*

*might end up drawing inaccurate conclusions such as the evening having the most intense IIF.*

L340: missing "to" ... however: "included" in addition to what else? (Also, in a revised manuscript, I hope "characterization of LIIF" would be well defined by now or a different formulation be used.)

We have now replaced "included" with "used".

Atmospheric relevance and applicability:

The 2nd paragraph could be shortened considerably.

Thank you for the comment. This paragraph has been condensed and now reads (L409-L414): *First, one should consider, which polarity data to use, assuming both exist.  Differences in the polarities could, for example, be used to derive insight into the growth mechanisms during LIIF or to study the effect of polarity on LIIF. At most other times, however use of data for only one polarity is preferable, which could for example be the case due to the desire for a more straightforward application and analysis. As discussed in Sect. 3.1, in this case in Hyytiälä the negative polarity would be a preferable choice.*

L405-407: This sentence ("Kulmala et al. (2020) recently ... particles and NPF.") is not useful. It does not seem to relate to the other parts of the text, while introducing new terms that are not explained. It is not necessary either. I suggest simply leaving it out.

Thank you for the suggestion. It has now been left out as suggested.

**Technical comments:**

L14: missing "a"

Thank you for noticing, it has been now added.

L109, 110: missing "the", "The"

Fixed.

L204: missing word?

Fixed (L235): *… during this time period.*

L209: "photogenic" should probably read "photosynthetic"

Thank you for the comment. Photogenic has been replaced with photochemical in the revised manuscript.

Fig. 1 caption: missing "a" ("a Neutral...") and "the" ("the SMEAR")

Also Figs. 2-5 and S1-S2. (And the missing "a" also in Fig. 7.)

Thank you for bringing this up. Fixed.

For all these figures I would also consider thicker lines and possibly different line styles. At the moment, the different styles are difficult to tell apart from each other.

Thank you for the comment. Thicker lines have been implemented.

Fig. 6 caption: "The growth rate"

Fixed.

Fig. 6: suggest horizontal grid lines for better orientation

Added.

---

## Referee Report (RR1)

Thank you for the careful consideration and implementation of my comments. I think the paper is ready for publication after a few technical corrections as listed below.

L34: "estimated to be over a half" -> "estimated to be over half",

L73-74: "do not tell us whether the clusters are growing or not in size" -> "do not tell us whether the clusters are growing or not".

L184-185: "Narrow size bin wide diameter ranges…" -> "Narrow diameter ranges" or something like that.

L342 The reference should be to Enghoff and Svensmark 2017 (Enghoff, M.B. and Svensmark, J., J. Aerosol Sci 114 p13-20, 2017) (and it should be added to the list of references).

L451: "precursor vapor, such as low volatile organic compounds, concentrations.", delete ", concentrations" -> "precursor vapors, such as low volatility organic compounds."